# Biological Degradation of the Azo Dye Basic Orange 2 by *Escherichia coli*: A Sustainable and Ecofriendly Approach for the Treatment of Textile Wastewater

Muhammad Ikram [1], Mohammad Naeem [1,*], Muhammad Zahoor [2,*], Marlia Mohd Hanafiah [3,4], Adeleke Abdulrahman Oyekanmi [3], Riaz Ullah [5], Dunia A. Al Farraj [6], Mohamed S. Elshikh [6], Ivar Zekker [7] and Naila Gulfam [8]

1 Department of Chemistry, Abdul Wali Khan University Mardan, Mardan 23200, Pakistan; ikrambiochem2014@gmail.com
2 Department of Biochemistry, University of Malakand at Chakdara, Chakdara 18800, Pakistan
3 Department of Earth Sciences and Environment, Faculty of Science and Technology, Universiti Kebangsaan Malaysia, Bangi 43600, Malaysia; mhmarlia@ukm.edu.my (M.M.H.); abdulkan2000@yahoo.com (A.A.O.)
4 Centre for Tropical Climate Change System, Institute of Climate Change, Universiti Kebangsaan Malaysia, Bangi 43600, Malaysia
5 Department of Pharmacy, College of Pharmacy, King Saud University, P.O. Box 2457, Riyadh 11451, Saudi Arabia; rullah@ksu.edu.sa
6 Department of Botany and Microbiology, College of Science, King Saud University, P.O. Box 2455, Riyadh 11451, Saudi Arabia; dfarraj@ksu.edu.sa (D.A.A.F.); melshikh@ksu.edu.sa (M.S.E.)
7 Institute of Chemistry, University of Tartu, 14a Ravila St., 50411 Tartu, Estonia; ivar.zekker@gmail.com
8 Department of Zoology, Jinnah College for Women, University of Peshawar, Peshawar 23000, Pakistan; nailazoo@yahoo.com
* Correspondence: naeem@awkum.edu.pk (M.N.); mohammadzahoorus@yahoo.com (M.Z.)

**Abstract:** In this study, initially 11 different bacterial strains were tested for the degradation capabilities against Basic Orange 2 dye. In initial screening with 78.90% degradation activity, *Escherichia coli* emerged as the most promising strain to degrade the selected dye, and was then employed in subsequent experiments. For further enhancing the degradation capability of selected bacteria, the effects of various physicochemical parameters were also evaluated. Among the tested parameters, 20 ppm dye concentration, 1666 mg/L glucose concentration, a temperature of 40 °C, 666 mg/L sodium chloride concentration, pH 7, 1000 mg/L urea concentration, a 3-day incubation period and the use of sodium benzoate as a redox mediator (666 mg/L) were found to be ideal conditions to get the highest decolorization/degradation activities. Finally, all the mentioned parameters were combined in a single set of experiments, and the decolorization capacity of the bacteria was enhanced to 89.88%. The effect of pH, dye concentration, incubation time and temperature were found to be responsible for the optimum degradation of dye ($p < 0.05$), as predicted from the ANOVA (analysis of variance) of the response surface methodology. The metabolites were collected after completion of the process and characterized through Fourier transform irradiation (FTIR) and gas chromatography mass spectrometry (GC-MS). From the data obtained, a proposed mechanism was deduced where it was assumed that the azo bond of the dye was broken by the azoreductase enzyme of the bacteria, resulting in the formation of aniline and 3, 4-diaminobezeminium chloride. The aniline was then further converted to benzene by deamination by the action of the bacterial deaminase enzyme. The benzene ring, after subsequent methylation, was transformed into o-xylene, while 3, 4-diaminobezeminium chloride was converted to p-xylene by enzymatic action. These findings suggest that *Escherichia coli* is a capable strain to be used in the bioremediation of textile effluents containing azo dyes. However, the selected bacterial strain may need to be further investigated for other dyes as well.

**Keywords:** biodegradation; Basic Orange 2; *Escherichia coli*; metabolites; azo dyes; wastewater

## 1. Introduction

Due to the increasing trend of urbanisation and industrialization, water pollution has emerged as one of the most important environmental challenges on a global scale [1]. Rapid industrialization and population increase, especially in developing countries, puts more strain on the availability of clean water resources and increases the issue of water quality [2]. Azo and anthraquinone dyes are the most widely used dyes worldwide due to their enormous range of colours [3]. Because of their versatility, cost-effectiveness, ease of use, high stability and colour intensity, azo dyes form the largest category (60–70%) of the overall synthetic dyes industry [4,5]. These dyes have a significant chromophore structure (-N=N), which ensures dye solubility in water and fibre adhesion [6,7]. The anthraquinone class is widely employed in the textile dyeing industry, with red dyestuff being particularly popular [8]. These dyes are well known for their water solubility, vibrant hues, and outstanding fastness [9]. Dyes are used for the coloration of several materials. These materials include food, cosmetics, textile fibers, tannery, pharmaceutical products, leather, paper, etc. [10,11]. The textile sector uses an estimated 80% of azo dyes for dying reasons, with about 10–15% of the dye not binding to the fibres during the process and thus being released into the environment [12]. Trace amounts of dye can lead to severe environmental and health hazards as some azo dyes are toxic, carcinogenic and mutagenic. Moreover, these hazardous dyes badly affect the distributions and compositions of aquatic animals, zooplanktonic and phytoplanktonic species, as well as aquatic ecosystems. [13,14]. The most prevalent hazardous pollutants are toxic chemicals and dyes in wastewater generated by heavy industries and other human activities, which must be treated before being discharged into the environment using cost-effective and ecologically friendly methods [15]. There are many reports on the use of physical or chemical treatment processes such as adsorption, chemical precipitation, photolysis, chemical oxidation and reduction, and electrochemical treatment for colour removal from dye-containing effluent [16]. However, these treatment approaches have some limitations, such as being commercially unviable and having limited applicability in entirely eliminating resistant azo dyes and their organic by-products [17]. Furthermore, additional pollution concerns could arise as a result of the creation of a large amount of sludge in some operations, such as chemical precipitation methods, etc., which require costly sludge removal operations [18]. Due to these obstacles, the development of biological approaches using a green chemistry approach is seen as a viable option because of its cost-effectiveness, environmental friendliness and public acceptability [19]. The development of cost-effective biological treatment technologies for industrial effluents is a major priority [20]. Bioremediation—using plants for the removal of dyes—is also a green approach. Recently, methylene blue dye (MBD) was eliminated from an aqueous solution using S. *latifolium* as an adsorbent [21]. The ability of *Lemna minor*, *Salvinia minima*, *Ipomoea aquatica*, and *Centella asiatica* as phytoremediation agents to remove pollutants from sewage wastewater samples has been also reported in a literature study [22]. Biobased modified absorbents are also effective for the removal of dyes and other polutants from wastewater. Nizam et al. reported [23] the effects of pre-treatment of graphite derived from agricultural wastes on the characteristics of graphene oxide for dye and heavy metal ion adsorption. Researchers and industries have become increasingly interested in bioremediation using microorganisms as the need for greener solutions has grown over time. Recently, many new approaches have been developed for the decolorization of dyes. One of these promising approaches of is the use of microorganisms to decolorize dyes. The dyes biodegradation is considered to be a cost-effective and environmentally friendly technique [24]. Both in laboratory and environmental samples, research has demonstrated that utilizing bacteria in the bioremediation of azo dyes in contaminated settings produces better results in the most diverse parameters and procedures [25]. Bacteria have emerged as promising options for dye decolorization procedures, as they break down organic contaminants/pollutants and use them as carbon and energy sources. Many bacterial strains, in particular, are highly tolerant of harmful contamination and have a short development cycle [26]. Development of a microbial ecosystem is also a green solution for the desulphurization of coal [27].

The first step in the bacterial degradation of azo dyes is the reductive breakdown of azo bonds, which produces potentially toxic aromatic amines, and the second step is the degradation of these aromatic amines. Although bacterial degradation can occur in the presence or absence of oxygen, aerobic mechanisms are almost solely responsible for the biodegradation of these amines. Given this, combining anaerobic and aerobic phases in the same process for the safest and most efficient eradication of environmental and human risk factors associated with these substances is the ideal strategy for treating azo-contaminated industrial waste [28]. The degradation of dye has been extensively reported using other physico-chemical methods in the literature. However, the use of bacterial degradation of dye has rarely been reported. Recently, Asad et al. [29] reported the degradation of azo dye methyl orange using *Pseudomonas aeruginosa*. In the literature, a wide range of anaerobic and aerobic bacterial strains such as *Pseudomonas* sp., *Micrococcus* sp., *Xenophilus* sp., *Acinetobacter* sp., *Geobacillus* sp., *Shewanella* sp., *Corneybaterium* sp., *Escherichia coli, Lactobacillus* sp., *Dermacoccus* sp., *Rhizobium* sp., *Proteus* sp., *Klebsiellla* sp., *Morganella* sp., *Rhabdobacter* sp., *Staphylococcus* sp., *Enterococcus* sp., *Bacillus subtilis, Clostridium* sp., *Alcaligenes* sp., *Aeromonas* sp. and *Alishewanella* sp. have been extensively reported in studies regarding biodegradation of azo dyes [30–33]. However, there are limited reports in the literature regarding Basic Orange 2 dye degradation using *Escherichia coli.* This study reports, for the first time, the biodegradation of Basic Orange 2 dye using *E. coli.* Additionally, the study critically examined the effect of process conditions both in one time factor optimization and optimization using central composite design to determine the effect of the interaction of the operational parameters on the degradation efficiency of *E. coli.*

The present study was aimed to select the best bacterial strain for the degradation of methyl red dye out of the available strains, and to understand the underlying mechanism of degradation. *Escherichia coli* was found to be the most efficient strain that effectively degraded the selected dye into o-xylene and p-xylene. Aniline and 3, 4-diaminobezeminium chloride were formed by the cleavage of the azo bond of the parent dye molecule by the azoreductase enzyme, and subsequent deamination lead to the formation of a benzene ring. The o-xylene was formed by the methylation of the benzene ring. Moreover, 3, 4-diaminobezeminium chloride was converted into top-xylene by subsequent deamination, followed by methylation of the benzene ring in the last step of the reaction. The effects of several physicochemical parameters were also investigated in order to get optimum dye degradation. Spectroscopic techniques such as UV-Vis, FTIR and GCMS were used for the analysis of dye degradation products.

## 2. Materials and Methods

### 2.1. Bacterial Strain Used

*Escherichia coli*, *Staphylococcus aureus*, *Staphylococcus epidermidis*, *Citrobacter amalonaticus*, *Bacillus subtilis*, *Xanthomonas campestris*, *Streptococcus pyogenes*, *Pseudomonas aeruginosa*, *Proteus mirabilis*, *Enterobacter sakazakii* and *Salmonella enterica* were initially tested for their biodegradation capability of the selected dye. The mentioned strains were obtained from the Department of Microbiology, Abdul Wali Khan University Mardan, Khyber Pakhtunkhwa, Pakistan.

### 2.2. Dye and Other Chemicals

The textile azo dye Basic Orange 2 (Figure 1) was obtained from the textile industry situated in Karachi, Pakistan. Nutrient broth, glucose, urea, sodium chloride, hydrochloric acid, sodium hydroxide, redox mediators and all other chemicals available were of the highest purity and analytical grade. The chemical structure of azo dye Basic Orange 2 is given below.

**Figure 1.** Chemical Structure of Basic Orange 2.

*2.3. Preparation of Dye Stock Solution*

First, 0.04 g of Basic Orange 2 dye was correctly weighted with the help of a digital balance. In a conical flask, this amount of dye was combined with a small amount of distilled water. After dissolving the dye, distilled water was added to bring the total volume to 1000 mL (40 ppm), while shaking continuously for 5 min. As a result, an aqueous stock solution of Basic Orange 2 with a concentration of 40 ppm was prepared and stored for future use.

*2.4. Growth of Bacteria*

Bacteria require growth media; thus, 13 g nutritional broth was dissolved in 1000 mL distilled water and shaken continuously for 5 min. The nutrient broth aqueous solution was prepared. To avoid any contamination and to kill unwanted microorganisms, the growth media nutrient broth, conical flasks, test tubes and all other glassware were sterilized at 121 °C for 3 h in an autoclave. Test tubes and nutrient broth solution were taken out from the autoclave and kept in a laminar flow hood to inoculate bacteria in medium. Each test tube was labeled and filled with 10 mL nutrient broth solution before being inoculated with bacterial biomass. After inoculation, the test tubes were kept in an incubator for 24 h at 37 °C to ensure the bacterial growth. After 24 h, the 5 mL dye solution was added to each test tube from the dye stock solution.

*2.5. Degradation/Decolorization Activity*

The absorbance of the supernatant was measured before decolorization. After 3 days, aliquots (5 mL) of the culture media were withdrawn and centrifuged at room temperature for 10 min at $10,000 \times g$ to separate the bacterial cell mass. The decolorization analysis was carried out using the supernatant recovered after centrifugation. In the visible area of a UV-Visible spectrophotometer, the absorbance of the supernatant extracted at different time intervals was measured at the dye's absorption maximum wavelength (498 nm). Using the following formula, the difference between the initial and final absorbance levels was used to calculate the percentage of degradation/decolorization [34].

$$\% \text{ Degradation} = \frac{\text{Initial absorbance} - \text{ final absorbance}}{\text{Initial absorbance}} \times 100 \qquad (1)$$

The bacterial strain that achieved the best decolonization rating was chosen for further degradation studies. *Escherichia coli* had the highest degradation/decolorization value (78.90%) in our experiment; hence, it was chosen and used for further dye decolorization (degradation) experiments.

*2.6. Optimization of Physiochemical Parameters for Biodegradation*

Optimal conditions are required for the dye to degrade properly. Physiochemical parameters such as dye concentration, pH, temperature, glucose concentration, time, sodium chloride concentration, urea concentration and redox mediators were studied in a variety of experiments. Three sets of experiments were carried out in each case. The detail of each set of experiments for the determination of optimum conditions is described below.

### 2.6.1. Effect of Dye Concentration on Degradation

To evaluate the effect of dye concentration on decolorization under static conditions, the selected bacterial culture was cultivated for 24 h in eight test tubes with 10 mL nutritional broth. The test tubes were filled with 5 mL of Basic Orange 2 solution from each concentration (5, 10, 15, 20, 25, 30, 35 and 40 ppm, respectively). For each concentration, eight control solutions were produced, each containing 10 mL nutritious broth and 5 mL Basic Orange 2 from stock solution. The culture tube combinations were spun at 10,000 rpm for 10 min at room temperature in a centrifuge machine after 3 days of incubation. The absorbance of the supernatant obtained after centrifugation was measured at 498 nm using a UV-Visible spectrophotometer [35].

### 2.6.2. Effect of Time on Dye Degradation

The inoculated nutrient broth media and dye solution were incubated in test tubes. The absorbance of the sample obtained after centrifugation was measured after three days. The percentage of decolorization was measured every 3 days, for 21 days.

### 2.6.3. Effect of Temperature on Dye Degradation

Six test tubes were filled with the inoculated sterile nutrient broth 10 mL solution. Each test tube was amended with 5 mL Basic Orange 2 from the stock solution (40 ppm). A control solution containing 10 mL nutrient broth and 5 mL methyl red was also prepared. After that, the tubes were incubated at 25, 30, 35, 40, 45, and 50 °C, respectively. The degraded samples in test tubes were centrifuged at 10,000 rpm for ten minutes at room temperature after a 3-day delay, then filtered using filter paper. In the same way as before, the percent decolorization was determined using a UV-Visible spectrophotometer.

### 2.6.4. Effect of pH on Dye Degradation

Bacteria require a specific pH for survival and growth; as such, sterile nutrient broth in test tubes were inoculated with the selected *Escherichia coli* culture and incubated at 37 °C. Control solutions containing 10 mL nutrient broth and 5 mL Basic Orange 2 solution were also prepared from the dye stock solution (40 ppm). The pH in control solutions, as well as in inoculated tubes, were adjusted using 1 M HCl and 1 M NaOH Solution. After a 3-day incubation period, the sample mixtures in test tubes were centrifuged at 10,000 rpm for 10 min in a centrifuge machine at room temperature, and the supernatant withdrawn was then filtered through filter paper. The percent decolorization was determined using the procedure described above using a UV-Visible spectrophotometer.

### 2.6.5. Effect of Glucose Concentration on Dye Degradation

Glucose is the main source of energy and acts as a carbon source for bacteria. Five test tubes containing 10 mL sterile nutrient broth were inoculated with a selected strain of bacteria. Then, 5, 10, 15, 20 and 25 mg of glucose and 5 mL Basic Orange 2 solution from stock solution (40 ppm) were added, respectively, to each test tube and incubated at 37 °C. Control solutions for each concentration of glucose containing 5 mL dye solutions were also prepared. The degraded samples in test tubes were centrifuged at 10,000 rpm for 10 min to separate the supernatant from the bacterial cell mass. The percentage decolorization of the supernatant was measured by a UV-Visible spectrophotometer in a similar method as discussed above in the formula.

### 2.6.6. Effect of Sodium Chloride Concentration on Dye Degradation

Sodium chloride is a significant salt that raises the salinity of sea water, affecting bacteria's ability to degrade dyes. Optimal saline conditions are usually required for pollutant degradation. Five test tubes with 10 mL of sterile nutrient broth were inoculated with the bacterial strain. These tubes were supplied with sodium chloride salt in amounts of 5, 10, 15, 20 and 25 mg/15 mL, respectively. Control solutions of each concentration were

also prepared. The percentage decolorization of the supernatant was determined by the formula given in Equation (1).

2.6.7. Effect of Urea Concentration (as Nitrogen Source) on Dye Degradation

A sufficient amount of urea, which also serves as a nitrogen source, is essential for bacterial development. In five test tubes containing 10 mL inoculated sterile nutritional broth and 5 mL Basic Orange 2 solution, 5, 10, 15, 20 and 25 mg of urea were added, respectively. Control dye solutions were also applied without being inoculated. After three days, the percentage of decolorization was evaluated using a UV-Visible spectrophotometer.

2.6.8. Effect of Various Redox Mediators on Dye Degradation

Bacteria used certain chemical substances as electron acceptors, and electron donors during electron transfer known as redox mediators. Dye degradation/decolorization is also an oxidation–reduction reaction. The reduction of the azo linkage of the azo dyes by bacteria strains involves reduction either anaerobically or aerobically. In this study, uric acid, hydroquinone, ethylenediaminetetraacetic acid (EDTA) and sodium benzoate were used as redox mediators.

Four test tubes containing 10 mL sterile nutrient broth and Basic Orange 2 dye solution inoculated with *Escherichia coli* had, respectively, uric acid, hydroquinone, ethylenediaminetetraacetic acid and sodium benzoate added, at a concentration of 1 mg/15 mL (66 mg/L). As a reference, four separate control solutions of each concentration were also prepared. The redox mediator with the highest decolorization value was chosen and employed in further decolorization (degradation) experiments.

*2.7. Dye Degradation at Optimum Conditions*

The effects of several physiochemical parameters such as pH, dye concentration, glucose, sodium chloride, urea, temperature, time and redox mediator concentration were evaluated as indicated above in order to determine the best conditions for decolorization. Following the determination of these optimum parameters, a decolorization experiment was conducted using these settings in single experiment. An optimization study was conducted using the optimum conditions of the most significant parameters to determine the synergistic interaction of process variables on the degradation of Basic Orange 2. This was achieved using the central composite design (CCD) of the response surface methodology (RSM).

*2.8. Extraction of the Formed Metabolites after Degradation*

The mixture containing the degraded dye products and bacterial mass at optimum conditions was crushed, centrifuged for 30 min at 10,000 rpm at room temperature, and then filtered. The supernatant of cell free culture was used for the extraction of metabolites. The supernatant was combined with an equal volume of ethyl acetate and agitated vigorously for 30 min. The aqueous and organic or ethyl acetate phases were separated using a separating funnel. The ethyl acetate was then evaporated at 40 °C to obtain a solid extract of the mixture.

2.8.1. Analysis of the Formed Metabolites of Basic Orange 2 Dye by GC-MS

To identify the metabolites formed after degradation, an Agilent USB-393752 gas chromatograph (Agilent Technologies, Palo Alto, CA, USA) with HHP-5MS 5% phenyl methyl siloxane capillary column (30 m × 0.25 mm × 0.25 μm film thickness; Restek, Bellefonte, PA, USA) equipped with an FID detector was used. The oven temperature was first maintained at 70 °C for 1 min, and then the temperature was raised to 180 °C in 5 min. Finally, the temperature of the machine was raised to 280 °C for 20 min. The detector temperature was 290 °C, while the injector temperature was 220 °C. Helium was used as the carrier gas at a flow rate of 1 mL/min). An amount of 1 μL of sample was injected in the split-less mode.

The metabolites were analyzed using the same GCMS system as before, but this time with an Agilent HP-5973 (Ramsey, Minneapolis, USA) mass selective detector in the electron impact mode (ionization energy: 70 eV) under the same operating parameters as before. Metabolites were found by comparing their retention times to those of previously reported compounds in the literature.

### 2.8.2. FTIR Analysis of the Formed Metabolites

The FTIR analysis of dye before and after treatment was performed by a Perkin Elmer Spectrum Two instrument (103385; Waltham, MA, USA).

### 3. Results and Discussions

*3.1. Most Efficient and Potential Bacterial Strain for Basic Orange 2 Degradation*

The decolorization potential of different bacterial strains is different. In our study, out of 11 bacterial strains, *Escherichia coli* emerged to be the most effective strain in terms of biodegradation potential of the selected dye. Its percentage decolorization potential was 78.90%. The decolorization of other bacterial strains is given in Figure 2.

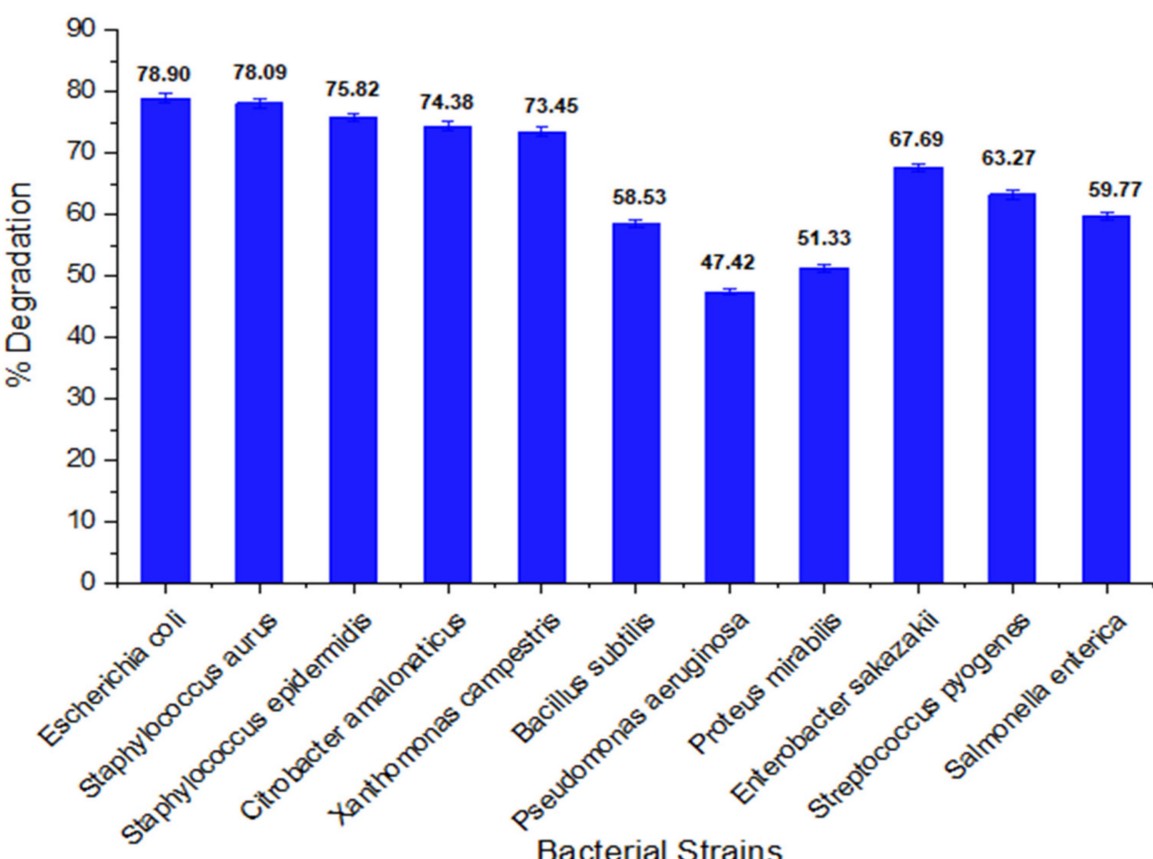

**Figure 2.** Percentage Degradation of Basic Orange 2 Dye by different Bacterial Strains.

*3.2. Dye Degradation at Optimum Conditions*

3.2.1. Effect of Dye Concentration on Degradation

The effect of dye concentration on the degradation rate is given in Figure 3. From the percentage degradation, it is clear that highest degradation (74.09%) is observed at 20 ppm, since dye is a chemical substance in which a higher concentration is toxic and limits the dye degradation activity of the bacteria. Zhuang et al., reported that a high concentration of the dye due to intrinsic toxicity blocks the active sites of enzyme, thus inhibits the bacterial activity and degradation potential [36]. Therefore, it can be stated that the dye degradation

rate increase when the dye concentration decreases, and decreases as the concentration of dye increases.

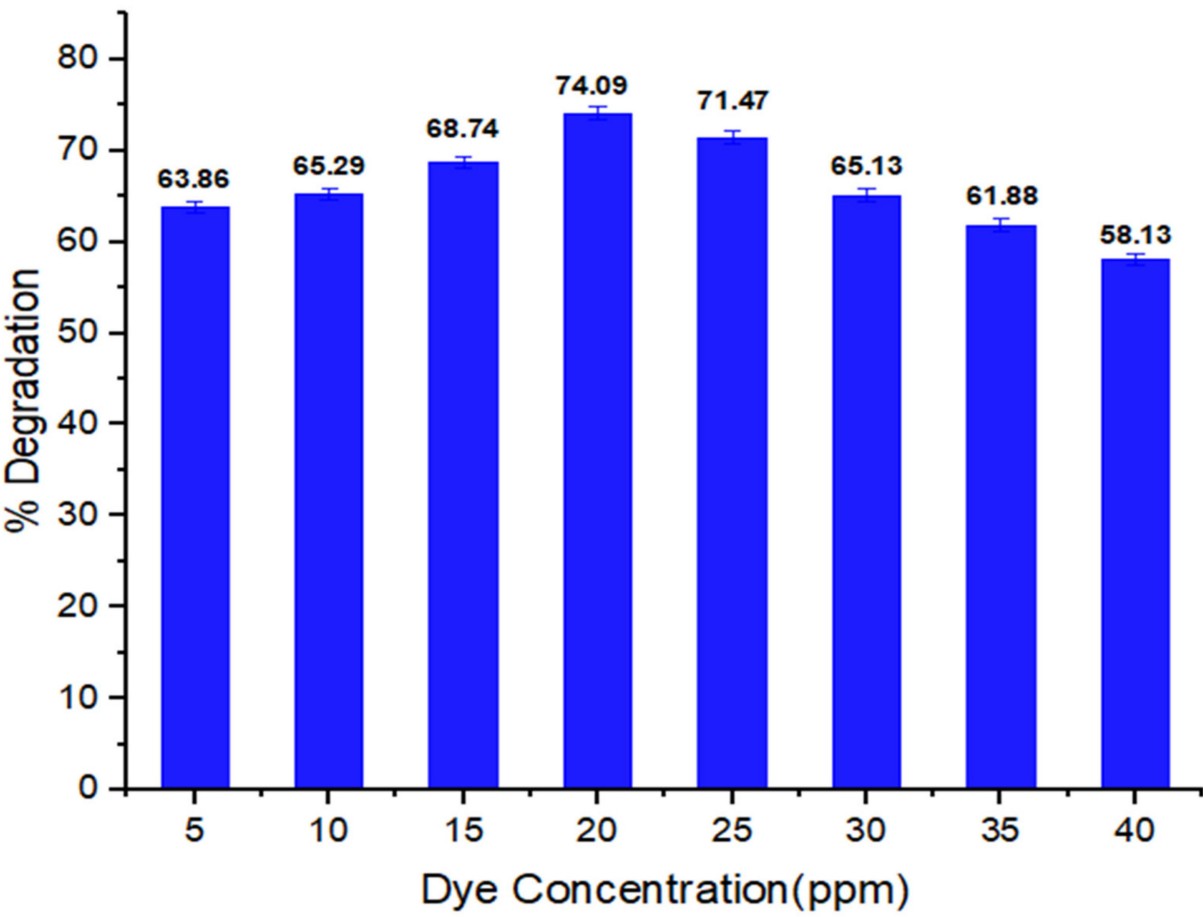

**Figure 3.** Effect of Dye Concentration on % degradation of Dye.

### 3.2.2. Effect of pH on Dye Degradation

One of the important factors that affects enzymatic activity and the biodegradation potential of bacteria is pH. An optimum pH is needed which results in a higher decolorization of the dyes. The effect of pH on the degradation of dye is shown in Figure 4. There is an increase in decolorization when pH increases. At pH 7, the decolorization activity is at the maximum (78.81%), and then a decrease occurs, which indicates that bacterial enzymatic activity and growth is affected at extreme acidic and alkaline conditions. For the most part, textile industrial processes occur in alkaline conditions, with the optimal pH range being 6 to 10 [37].

### 3.2.3. Effect of Temperature on Dye Degradation

The dye biodegradation potential of bacteria is affected by temperature. The effect of temperature on degradation is shown in Figure 5. Bacterial growth is affected by temperature, and consequently the biodegradation process is affected. At 40 °C, the highest decolorization (85.18%) was observed, which means that, below or above this temperature, the decolorization potential of *Escherichia coli* declines due to slow growth. Anjaneya et al. [38] reported that high temperature results in inactivation of the bacterial enzymes, and consequently the rate of decolorization of the bacteria decreases considerably. According to Pearce et al. [39], the optimum growth temperature ranges from 35 to 45 °C for the decolorization of reactive azo dyes until a maximum point. A steady decrease in decolorization occurs above the optimum temperature, which is probably due to enzyme denaturation.

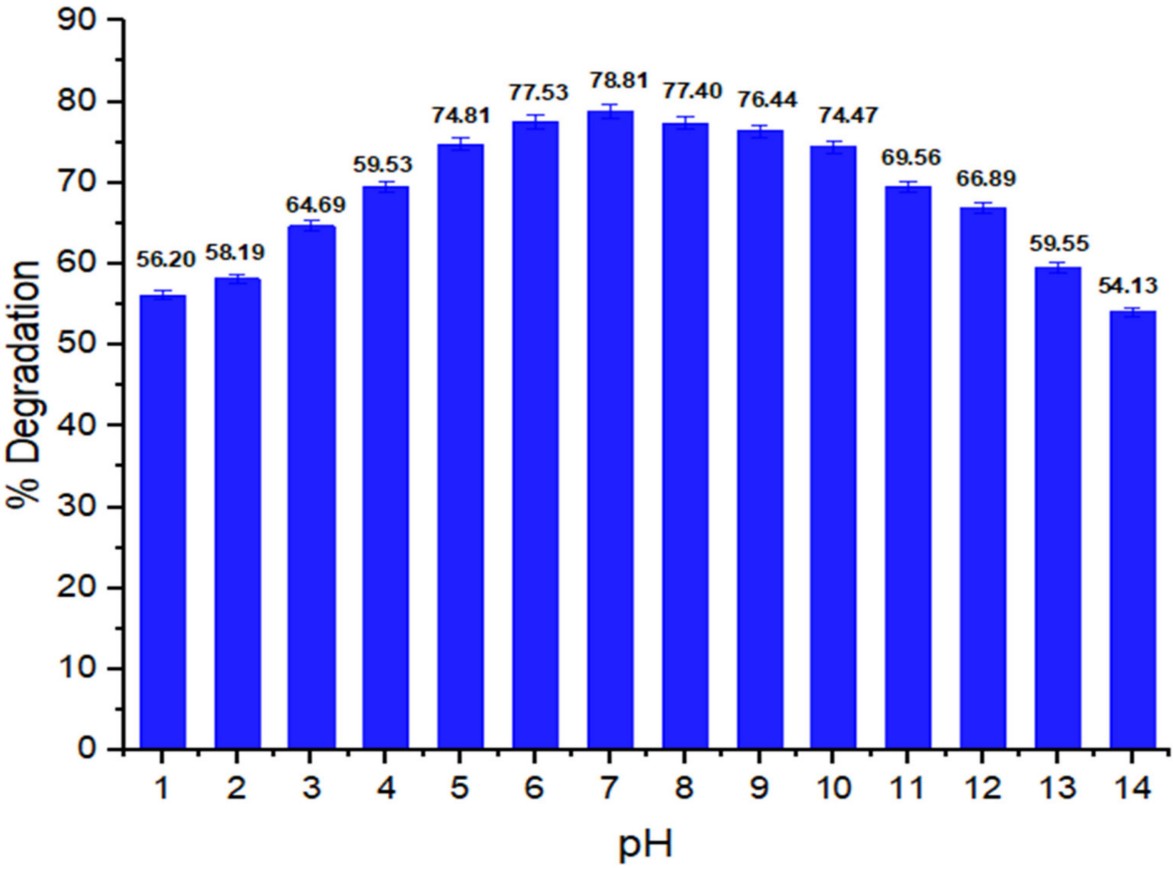

**Figure 4.** Effect of pH on % degradation of Dye.

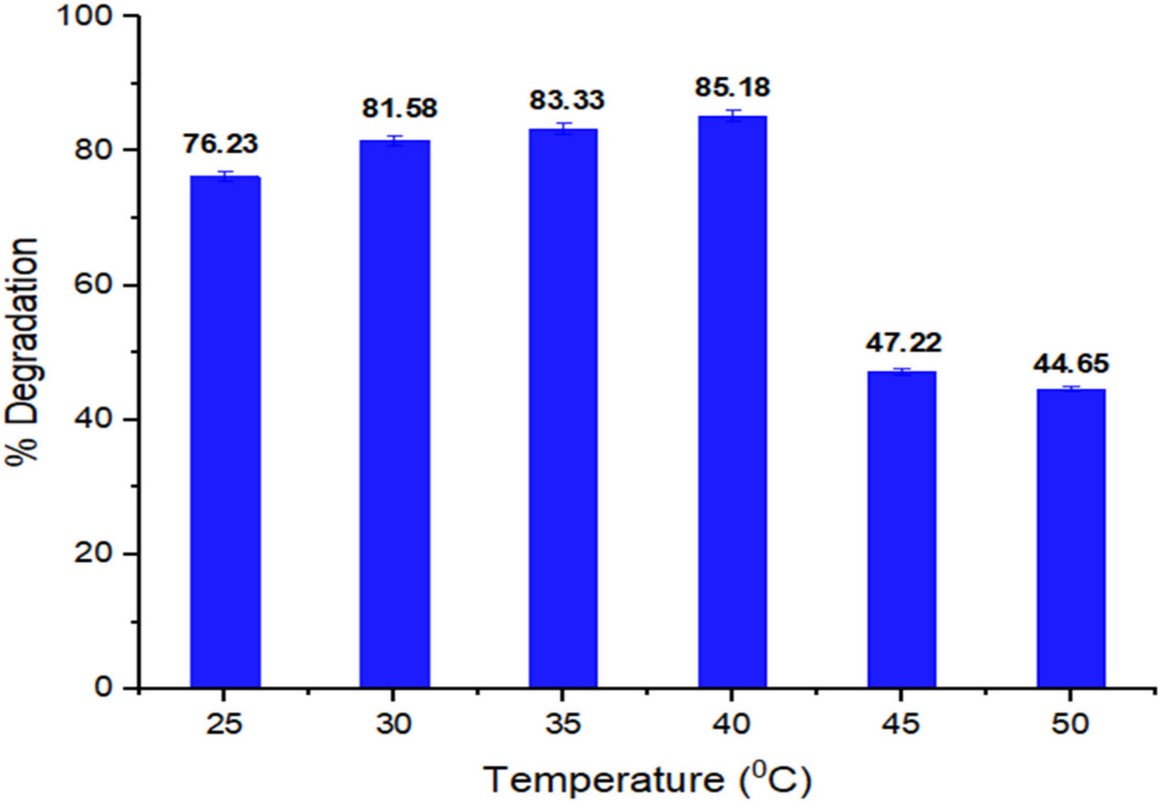

**Figure 5.** Effect of Temperature on % degradation of Basic Orange 2 Dye.

### 3.2.4. Effect of Incubation Time on Dye Degradation

The effect of incubation time on Basic Orange 2 degradation by *Escherichia coli* is shown in Figure 6. The dye degradation was recorded in 3-day intervals, up to 21 days. The maximum decolorization (82.92%) was observed after 3 days. The percentage decolorization decreased after a 3-day incubation period. Bacteria need growth media and other nutritional requirements, such as C, N, etc.; indeed, proper degradation is expected to begin when bacteria is alive and active. Bacterial enzymes are also responsible for degradation kinetics. The biomass production is rapid at first, but after three days, the number of bacterial colonies competing for nutrients diminishes. The acclimatization period began after 24 h, because after this period bacteria were fully grown and active. The maximum degradation kinetics occurred at 72 h. After this time, degradation decreased, which is largely due to bacterial enzyme saturation and a decrease in growth.

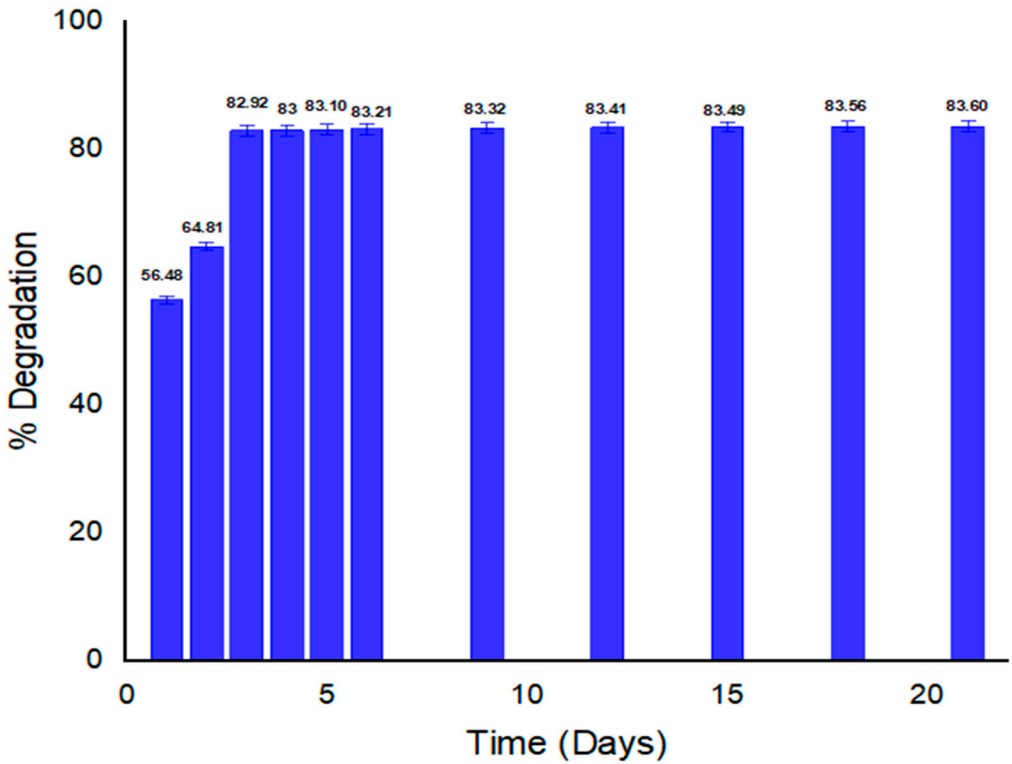

**Figure 6.** Effect of Incubation Time (Days) on Basic Orange 2 Dye % degradation.

### 3.2.5. Effect of Glucose Concentration Dye Degradation

Glucose serves as a carbon source for bacteria; therefore, a sufficient amount of glucose is needed for bacterial biomass production and the breakdown of dye. In Figure 7, the glucose effect on the degradation of the selected dye is shown. A glucose concentration of 1666 mg/L showed a high degradation rate (89.71%). Some complex dyes are difficult to degrade, and as such, additional carbon is needed to supplement carbon [40].

### 3.2.6. Effect of Urea Concentration Dye Degradation

As bacteria use urea as a nitrogen source, a substantial amount of urea is required for bacteria to degrade the dye. Figure 8 depicts the impact of urea concentration on the degradation of the selected dye. The dye showed a high degradation rate (80.92%) at a concentration of 1000/L. At higher concentrations, the breakdown activity was reduced due to urea toxicity, as the content of urea had increased.

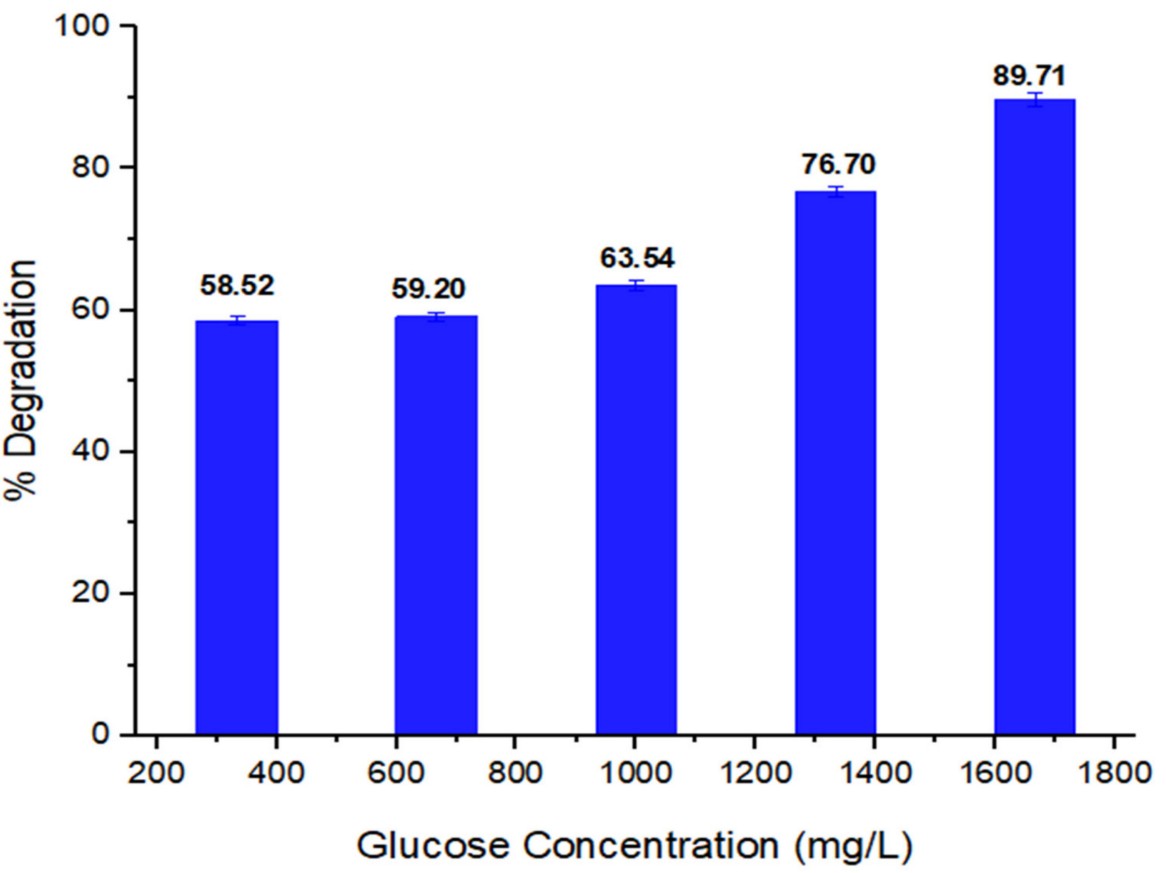

**Figure 7.** Effect of Glucose Concentration on Basic Orange 2 Dye % degradation.

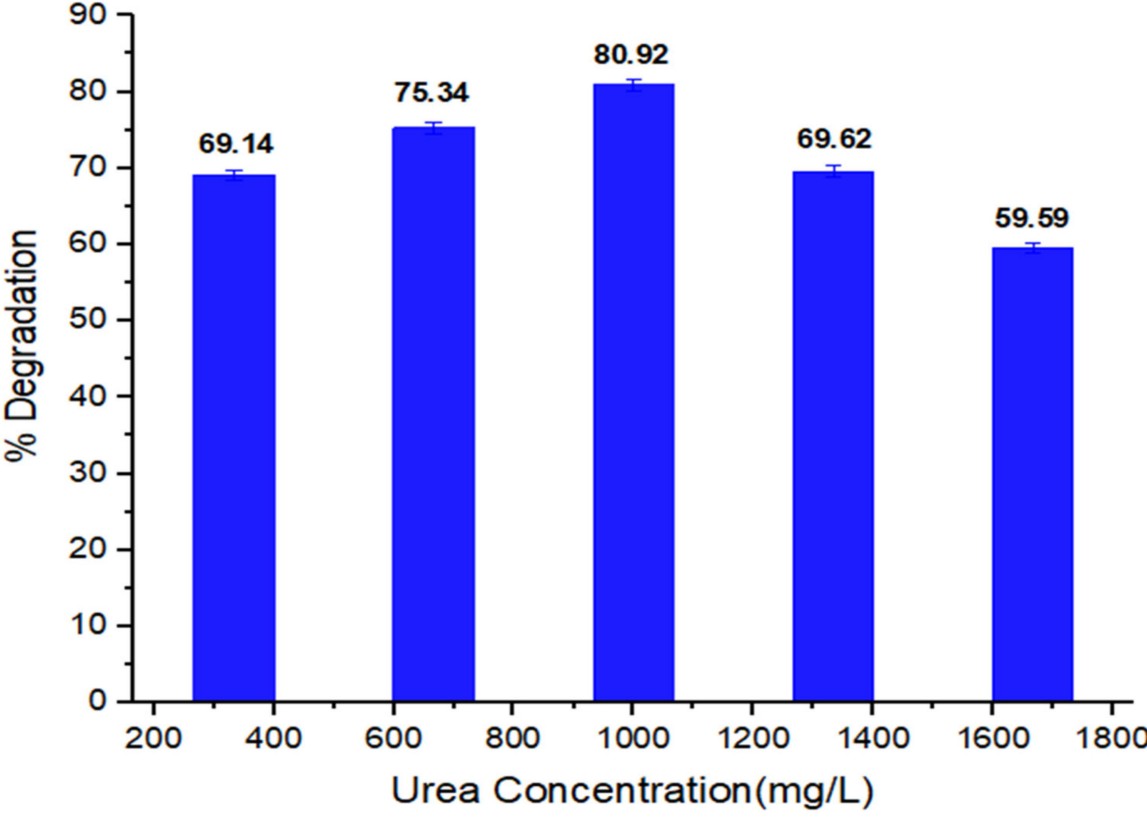

**Figure 8.** Effect of Urea Concentration (mg/L) on Basic Orange 2 Dye % degradation.

### 3.2.7. Effect of Sodium Chloride Concentration on Dye Degradation

Figure 9 depicts the effect of sodium chloride supplementation on dye degradation. As the concentration rises, the bacterial ability to degrade dye decreases. The percentage degradation at 666 mg/L is 82.35, indicating that an optimal salt concentration is required. Plasmolysis of bacterial cells is induced by high salt concentrations, which reduces bacterial growth, and as a result, its dye degradation capability decreases [41]. Asad et al., also found a percentage degradation of 41.23 for Dye Brown 706 biodegradation at low NaCl concentrations (0.1 gm/15 mL) [42].

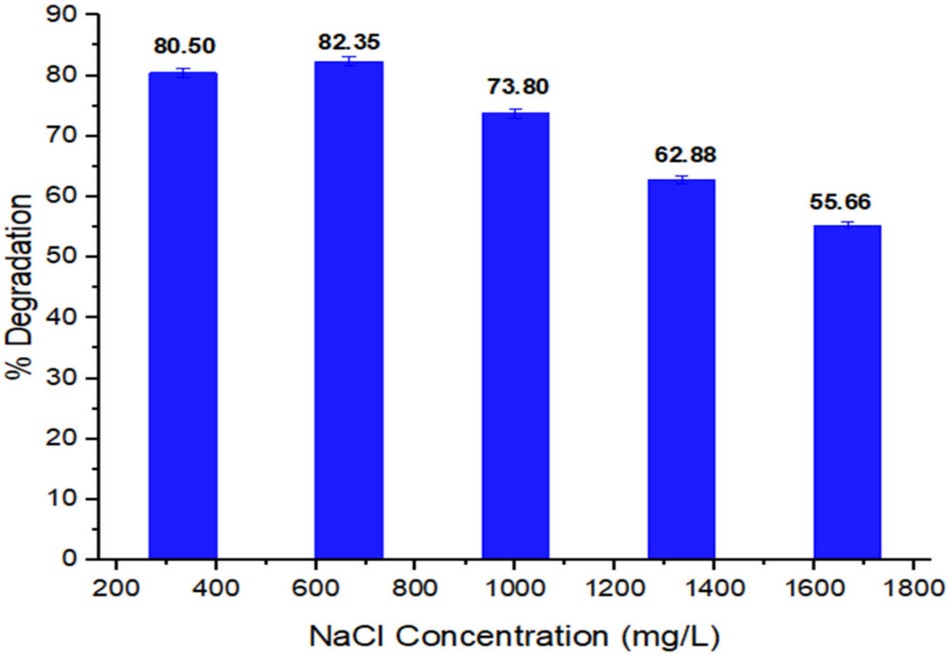

**Figure 9.** Effect of Sodium Chloride Concentration (mg/L) on Dye Basic Orange 2% degradation.

### 3.2.8. Effect of Redox Mediators on Dye Degradation

During electron transfer, bacteria utilize redox mediators, which are chemical compounds that act as electron acceptors and donors. Dye decolorization is an oxidation–reduction reaction. The azo bond of the dyes is reduced by bacteria strains either anaerobically or aerobically. Figure 10 shows the effect of redox mediators at a concentration of 66 mg/L on dye degradation; sodium benzoate has the greatest degradation capability on the dye (81.19%). Our results are remarkably comparable with a previously published study where the highest degradation was observed by using a redox mediator. The transference of reducing equivalents from the primary electron donor (such as cosubstrate) to the terminal electron acceptor (such as azo dye) has been shown to be a rate-limiting step in the anaerobic azo dye reduction process [43]. Slight redox mediator supplementation is sufficient for speeding up the electron transfer phase while reducing dye molecule steric hindrance [44,45]. In other experiments, the addition of hydroquinone as a mediator resulted in a 94.41 percent yield after 5 h of incubation, compared to 79.35 percent in the control experiment (dye in nutritional broth without any mediator) [46].

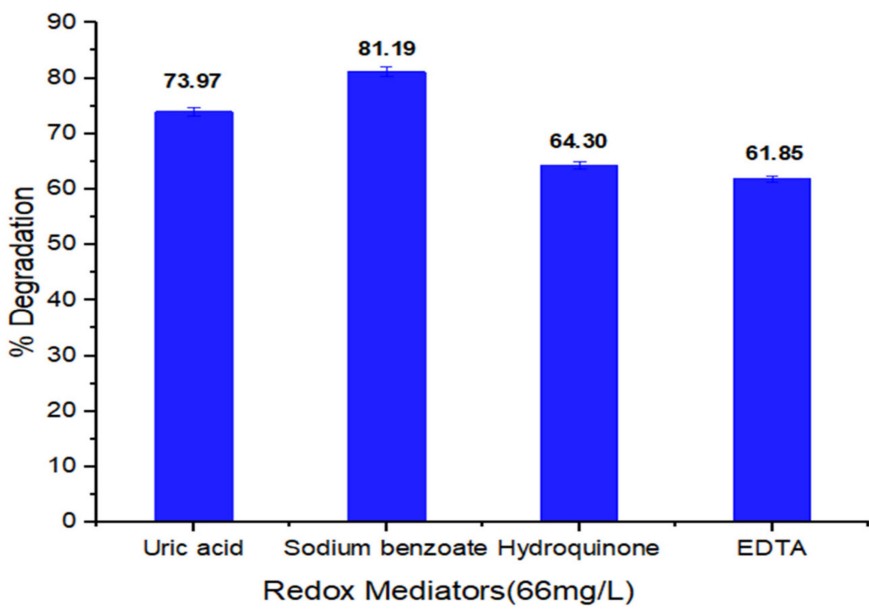

**Figure 10.** Effect of Redox Mediators (mg/L) on Basic Orange 2 Dye % degradation.

*3.3. Degradation of Dye Basic Orange 2 at Optimum Physiochemical Conditions*

After determining the effects of various physicochemical parameters on dye degradation by selected bacteria, such as dye concentration, pH, temperature, sugar concentration, urea concentration, redox mediators, salt concentration and time, the degradation of the selected dye was conducted by applying these optimum conditions in a single experiment. The optimum conditions found were 20 ppm Basic Orange 2 concentration, pH 7, a temperature of 40 °C, 1666 mg/L glucose concentration, 666 mg/L sodium chloride concentration, 1000 mg/L urea concentration, 3-day incubation duration and 66 mg/L sodium benzoate as a redox mediator. The other experimental conditions were the same as those mentioned previously. After combining these optimum conditions in a single final experiment, 89.88% degradation was achieved. Figure 11 shows dye Basic Orange 2 color before and after bacterial treatment. The influence of one factor at a time on the degradation of the dye cannot determine the interactive effects among the process variables studied [47]. As a consequence, this optimization technique does not describe the complete effects of the variable on the response.

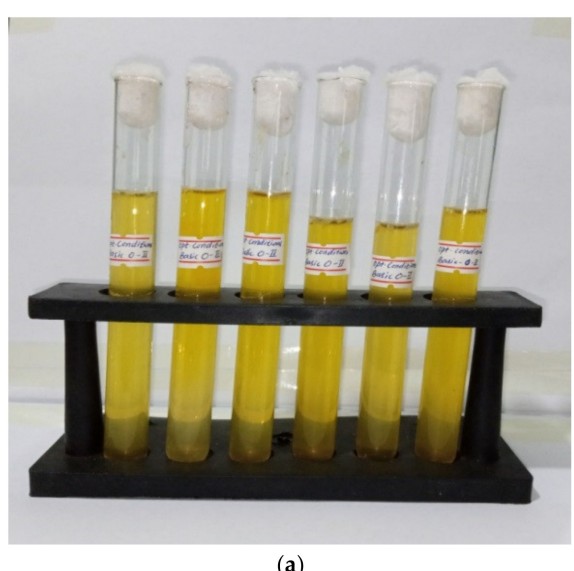

(**a**)

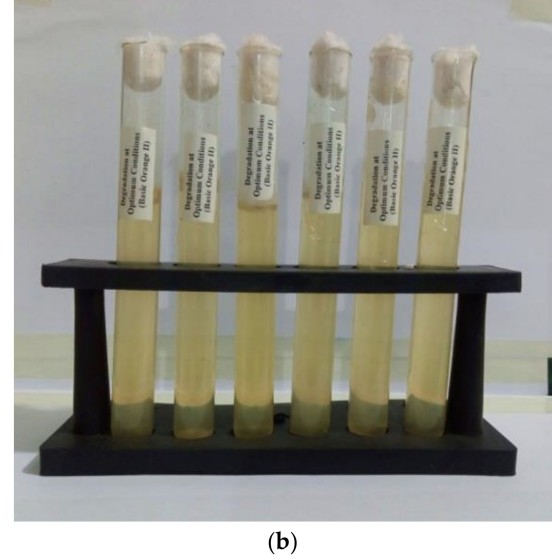

(**b**)

**Figure 11.** (**a**) Basic Orange 2 before Treatment of *E. coli*. (**b**) Basic Orange 2 after *E. coli* Treatment.

### 3.4. Response Surface Optimization of Most Significant Parameters

The experimental design evaluating the effect of the process conditions for the optimization of the dye degradation is presented in Table 1, which is evaluated according to Equation (1). A close agreement between the observed values and the predicted values indicates the suitability of the model.

$$Y = \beta_0 + \sum_{I=1}^{K} \beta_I X_i + \sum_{I=1}^{K} \beta_{II} X_i^2 + \sum\sum_{I>j}^{K} \beta_{Ij} X_1 X_j + \text{\euro} \tag{2}$$

where $\beta_0$ and $\beta_i$ denote the constant coefficient; $X_i$ represents the linear coefficient of the input parameters; $\beta_{ii}$ signifies the quadratic coefficient of the input parameter, $X_i$; $\beta_{ij}$ describe the interaction coefficient between the input parameters $X_i$ and $X_j$; and $\text{\euro}$ indicates the error of the model.

**Table 1.** Analysis of central composite design for the degradation of Basic Orange 2 dye.

| Factors | Units | Code | Levels | | |
|---|---|---|---|---|---|
| | | | −1 | 0 | +1 |
| pH | - | $X_1$ | 3 | 6.5 | 10 |
| Dye concentration | ppm | $X_2$ | 5 | 12.5 | 20 |
| Incubation time | day | $X_3$ | 3 | 12 | 21 |
| Temperature | °C | $X_4$ | 25 | 37.5 | 50 |

The relationship between the actual value and the predicted result of the degradation of Basic Orange 2, based on the quadratic function of the second order model, is represented in coded form in Equation (2).

$$Y = 83.40 + 5.11X_1 - 2.70X_2 - 0.083X_3 + 2.13X_4 - 16.84X_1^2 - 3.31X_2^2 - 3.38X_3^2 - 2.58X_4^2 + 2.17X_1X_2$$
$$-4.01X_1X_3 + 0.91X_1X_4 + 4.10X_2X_3 + 1.05X_2X_4 - 1.09X_3X_4 \tag{3}$$

The result of the analysis of variance (ANOVA) is presented in Table 2. The low value of probability ($p < 0.05$) and high F-values indicated the suitability of the model. The high correlation coefficient ($R^2 > 0.92$) indicated good agreement between the observed and the predicted values, as observed in the adjusted $R^2$. Adequate (Adeq) precision gives an indicator of the signal to noise ratio. The present ratio of 15.585 > 4 revealed that the model exhibited an adequate signal to navigate the design space [48].

The interaction between the process variables to determine the synergistic effect of the degradation of Basic Orange 2 dye using *Escherichia coli* is illustrated in Figure 12a–d. All response surface plots revealed that the process variables of pH, dye concentration, incubation time and temperature significantly enhanced the degradation of Basic Orange 2 dye. It is clear that an increase in dye concentration and pH induced increased degradation efficiency of *Escherichia coli* (Figure 12a). This phenomenon is due to the initial dye concentration providing the driving force to overcome the resistance barrier to the mass transfer of dye molecules between the aqueous phase and the solid phase. The increase in initial dye concentration therefore enhances the interaction of Basic Orange 2 and *Escherichia coli* [49]. By contrast, *Escherichia coli* has the ability to facilitate faster degradation of Basic Orange 2 dye; this is because a rapid reaction rate was achieved at the initial stage ($p < 0.05$) in neutral pH conditions (Figure 12b). The interaction of temperature and pH optimally influenced the degradation of Basic Orange 2 as the temperature and pH increased (Figure 12c). Temperature (>37 °C) is the main factor limiting the enzymatic activities and stability of *Escherichia coli* [50]. An increase in incubation time and dye concentration favours the degradation of the dye (Figure 12d), although an increase in dye concentration beyond 12.5 ppm may have an inhibitory effect on bacterial growth and its metabolic activities [51]. The results revealed that under process conditions of the investigated parameters, optimum degradation of the dye was achieved at a pH of 6.94, a dye concentration of 19.89 ppm,

within 10.79 incubation days at 47.97 °C. Under these conditions, the interaction of the process factors, as indicated in Figure 13, achieved the optimum degradation of Basic Orange 2 dye by *Escherichia coli*. Table 3 shows ANOVA of percentage degradation of Basic Orange 2 dye after bacterial treatment.

**Table 2.** Central composite design matrix.

| Run | $X_1$ | $X_2$ | $X_3$ | $X_4$ | Actual Value | Predicted Value |
|-----|-------|-------|-------|-------|--------------|-----------------|
| 1 | 0 | 0 | 2 | 0 | 52.10 | 59.97 |
| 2 | −1 | −1 | −1 | −1 | 58.20 | 68.04 |
| 3 | 1 | −1 | 1 | 1 | 31.40 | 35.92 |
| 4 | 0 | −2 | 0 | 0 | 52.30 | 56.69 |
| 5 | −1 | −1 | −1 | −1 | 56.60 | 57.80 |
| 6 | −1 | 1 | 1 | −1 | 56.40 | 53.82 |
| 7 | 1 | 1 | 1 | −1 | 48.90 | 54.15 |
| 8 | 0 | 0 | 0 | 2 | 52.70 | 58.87 |
| 9 | −1 | 1 | −1 | −1 | 56.10 | 58.49 |
| 10 | 0 | 0 | 0 | 0 | 78.30 | 74.20 |
| 11 | 0 | 0 | 0 | 0 | 38.90 | 42.60 |
| 12 | 0 | 2 | 0 | 0 | 59.70 | 67.05 |
| 13 | −1 | −1 | 1 | −1 | 59.20 | 55.97 |
| 14 | 0 | 0 | 0 | −2 | 51.60 | 55.67 |
| 15 | 0 | 0 | −2 | 0 | 57.80 | 56.52 |
| 16 | 0 | 0 | 0 | 0 | 67.60 | 64.89 |
| 17 | 1 | −1 | 1 | −1 | 9.20 | 5.83 |
| 18 | −1 | 1 | −1 | 1 | 32.60 | 26.26 |
| 19 | −2 | 0 | 0 | 0 | 76.40 | 75.54 |
| 20 | −1 | 1 | 1 | 1 | 73.60 | 64.74 |
| 21 | 1 | −1 | −1 | −1 | 81.20 | 70.06 |
| 22 | 1 | 1 | −1 | −1 | 68.30 | 69.72 |
| 23 | 1 | 1 | −1 | 1 | 80.30 | 68.82 |
| 24 | 1 | −1 | −1 | 1 | 75.60 | 77.36 |
| 25 | 0 | 0 | 0 | 0 | 83.40 | 83.40 |
| 26 | 1 | 1 | 1 | 1 | 83.40 | 83.40 |
| 27 | 0 | 0 | 0 | 0 | 83.40 | 83.40 |
| 28 | 2 | 0 | 0 | 0 | 83.40 | 83.40 |
| 29 | −1 | −1 | 1 | 1 | 83.40 | 83.40 |
| 30 | 0 | 0 | 0 | 0 | 83.40 | 83.40 |

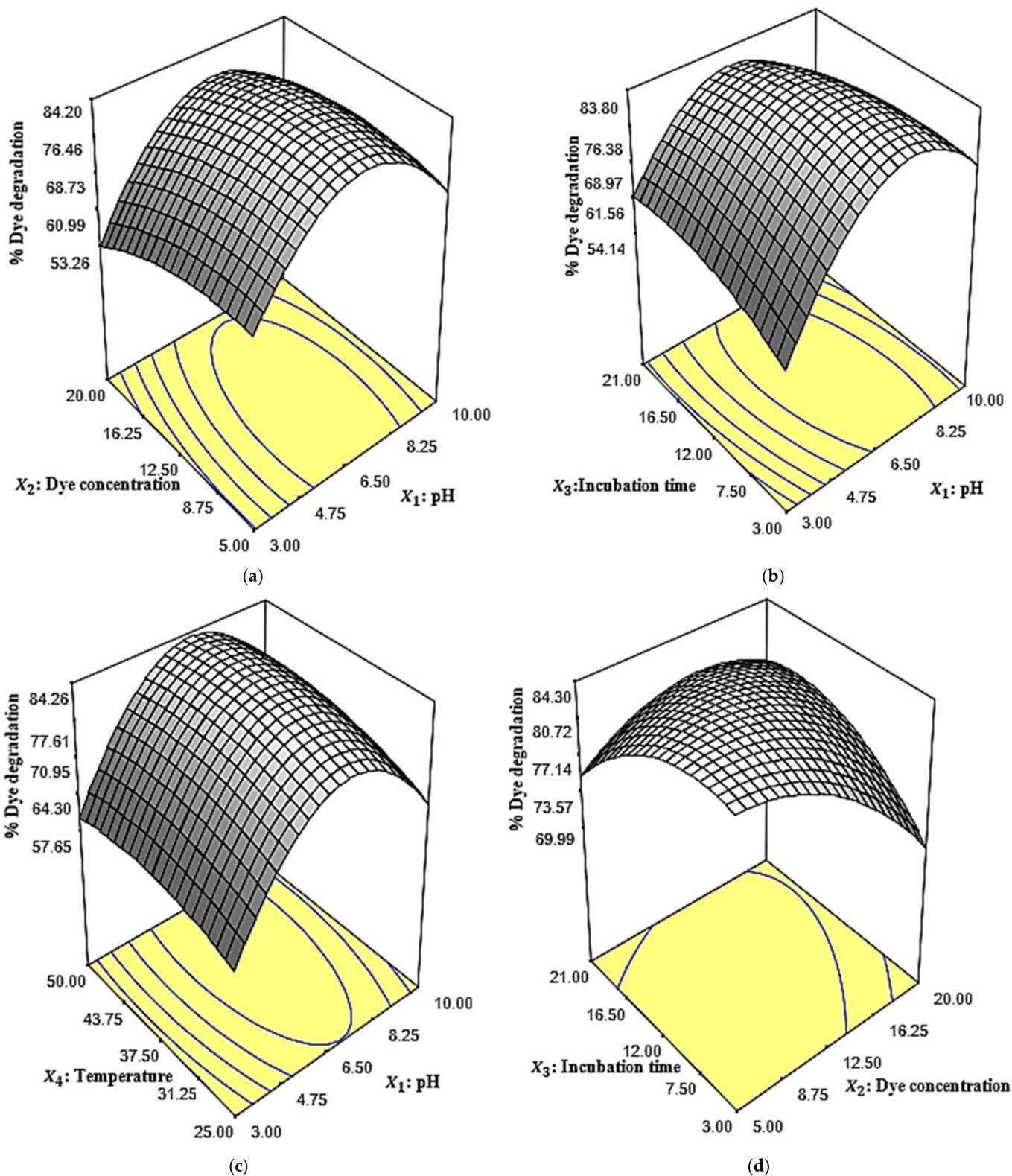

**Figure 12.** 3D surface plot for the interaction of (**a**) pH and dye concentration, (**b**) pH and incubation time, (**c**) pH and temperature, and (**d**) incubation time and dye concentration, for the degradation of Basic Orange 2 dye using *Escherichia coli*.

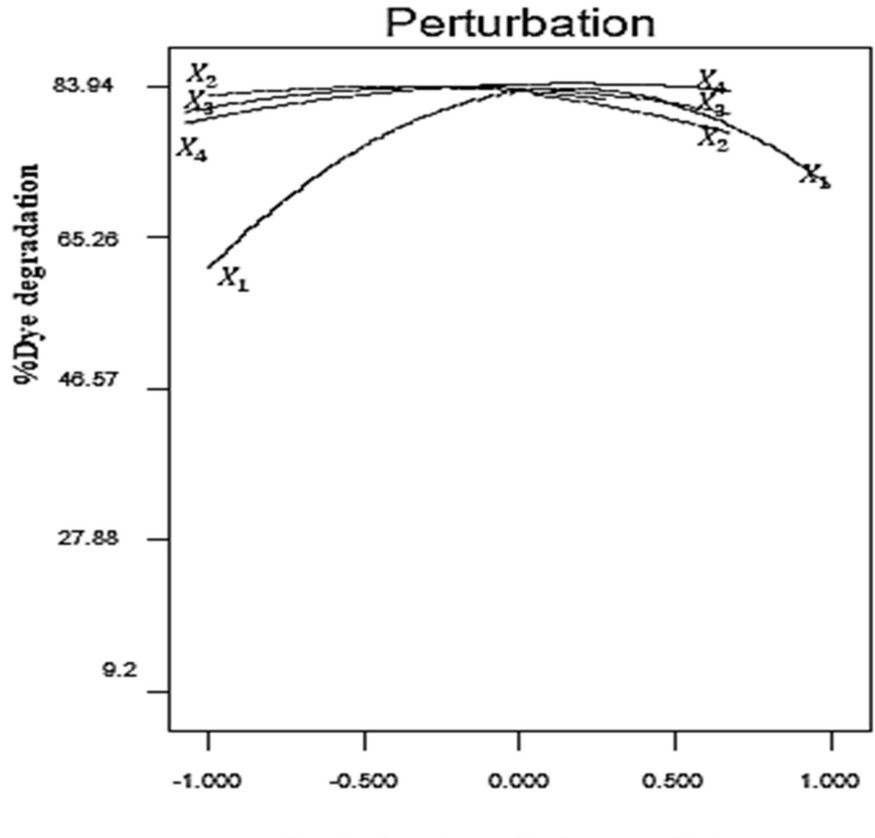

**Figure 13.** Optimum percentage degradation of Basic Orange 2 dye under the interaction of process variables.

**Table 3.** ANOVA of percentage degradation of Basic Orange 2 dye.

| Source | Sum of Squares | DF | Square Values | F-Value | *p*-Value | Remarks |
|---|---|---|---|---|---|---|
| Model | 9384.84 | 14 | 670.35 | 13.53 | <0.0001 | Significant |
| $X_1$ | 626.28 | 1 | 626.28 | 12.64 | 0.0029 | Significant |
| $X_1{}^2$ | 7777.96 | 1 | 156.97 | 156.97 | <0.0001 | Significant |
| $X_2{}^2$ | 301.34 | 1 | 301.34 | 6.08 | 0.0262 | Significant |
| $X_3{}^2$ | 312.81 | 1 | 312.81 | 6.31 | 0.0239 | Significant |
| $X_1 X_3$ | 257.60 | 1 | 257.60 | 5.20 | 0.037 | Significant |
| $X_2 X_3$ | 268.96 | 1 | 268.96 | 5.43 | 0.0342 | Significant |

| R-Squared | Adj R-Squared | Adeq Precision | Mean | PRESS |
|---|---|---|---|---|
| 0.9266 | 0.8581 | 15.585 | 62.51 | 4281.04 |

*3.5. Characterization Study*

The characterization of Basic Orange 2 dye metabolites was analyzed using Fourier transform infrared (FT-IR) and gas chromatography and mass spectrometry (GC-MS) analysis.

3.5.1. Fourier-Transform Infrared (FTIR) Analysis

The original dye and metabolites of the optimum set of experiments were characterized by FTIR. The FTIR spectrum of degraded dye was compared with that of the original non-degraded dye. FTIR spectra indicated that certain specific functional groups present in the original dye were totally absent in the degraded products, and new peaks appeared.

The FTIR spectra of Basic Orange 2 is given in Figure 14a. The peaks at 3289.55 cm$^{-1}$ and 3128.7 cm$^{-1}$ represent the N-H stretch of amine. The peak at 1620.14 cm$^{-1}$ represents N=N stretching or the azo bond of the dye molecule. The peak at 1254.61 cm$^{-1}$ indicates C-H stretching. At 821.38 cm$^{-1}$, there is a peak for N-H stretching of the aromatic amine ring. The FTIR spectra of degraded dye is also provided in Figure 14b. In comparison to the first spectra, the peaks at 3289.55 cm$^{-1}$ and 3128.7 cm$^{-1}$ have disappeared. The new peaks at 2927 cm$^{-1}$ and 2857 cm$^{-1}$ represent C=C and C-H stretching, respectively. The peak at 1620.14 cm$^{-1}$ for N=N has disappeared, which means that the azo bond was reduced and the dye was degraded by the bacterial enzyme azoreductase. Moreover, the peak at 1715 cm$^{-1}$ represents C-H stretching. The new peak at 667 cm$^{-1}$ represents the C=C stretch of the benzene ring. When comparing the FTIR spectra of the original dye to the spectra of the degraded dye metabolites, significant differences can be noticed. Some peaks have disappeared, while others have appeared, indicating that the dye has been decomposed and new molecules or metabolites have formed.

### 3.5.2. Gas Chromatography and Mass Spectrometry

Figure 15a,b shows the GC and GC-MS chromatograms of the metabolites, respectively. The dye metabolite found at RT 2.24, with a charge to ion mass (m/z) that closely relates to the dye structure, was identified as o-xylene and p-xylene, respectively. Some other compounds were formed, but their degradation mechanism is unknown. Because commercial grade solvents were used in the extraction process, the majority of the compounds that may be present are solvent-based.

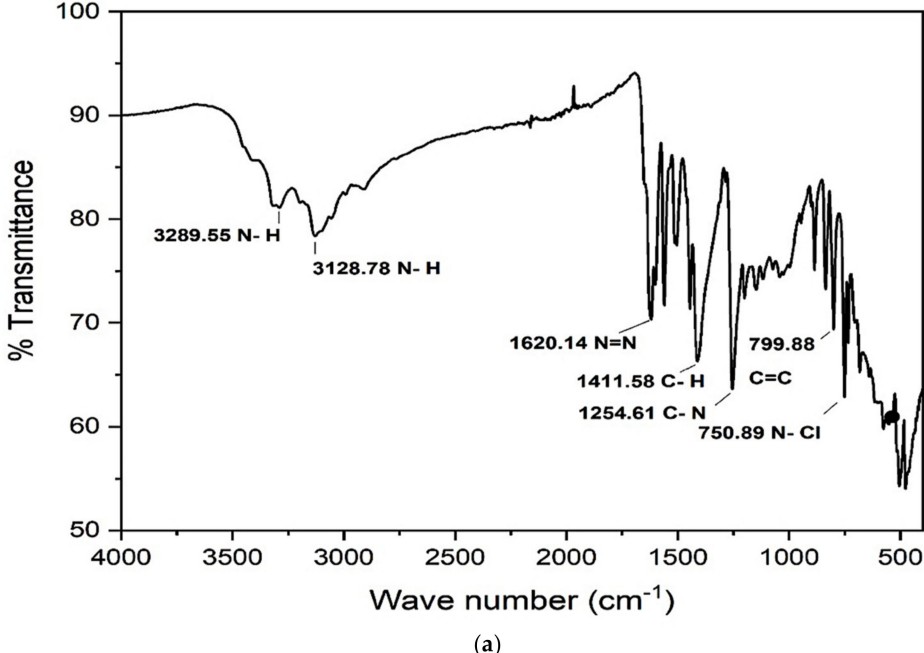

(**a**)

**Figure 14.** *Cont.*

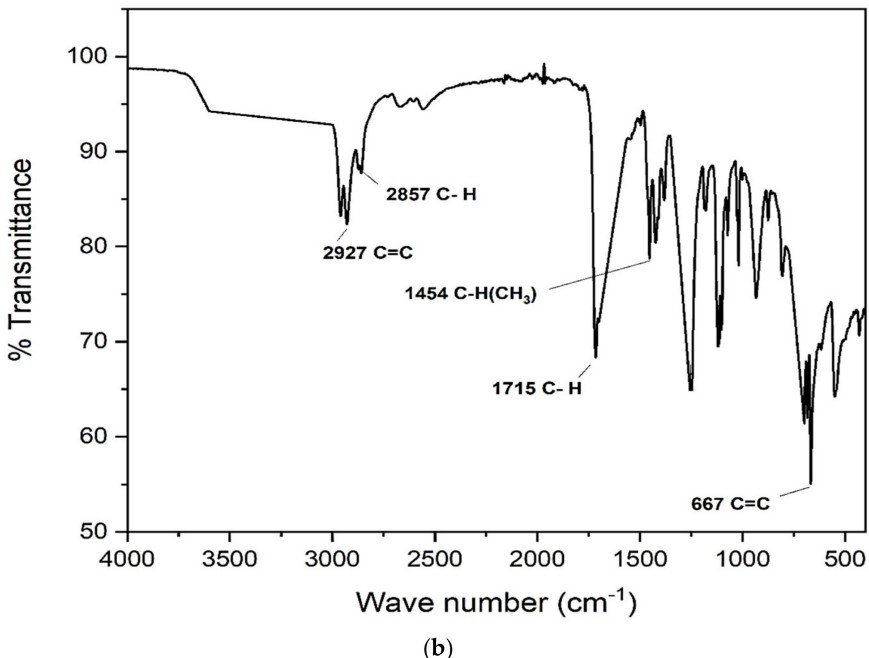

(b)

**Figure 14.** (**a**) FTIR Spectra of Dye Basic Orange 2 before Degradation. (**b**) FTIR Spectra of Basic Orange 2 Dye after Degradation by *Escherichia coli*.

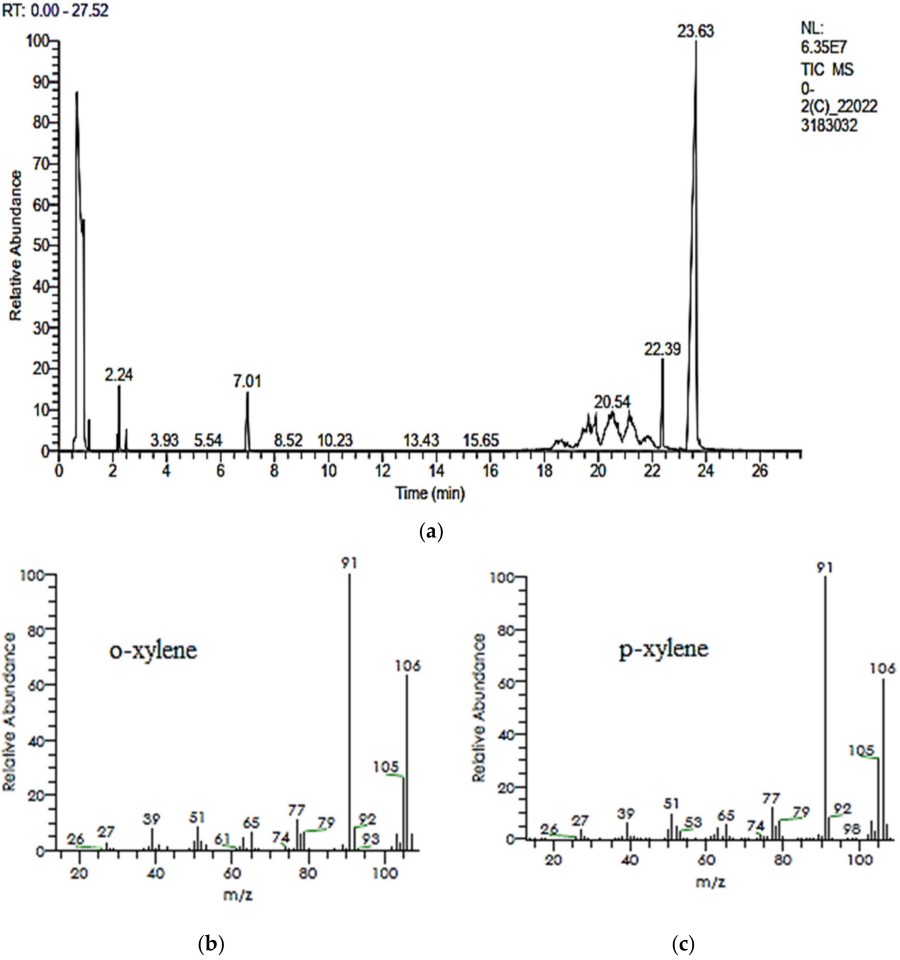

**Figure 15.** GC chromatogram of Basic Orange 2 dye: (**a**) after bacterial degradation; (**b,c**) major metabolites.

### 3.6. Proposed Mechanism for the Biodegradation of Basic Orange Dye by Escherichia coli

The azo dyes are broken down by enzymes such as azoreductase, peroxidase and laccase in the bacterial system. Azoreductase breaks the azo bond of the dye and can degrade azo dye in both aerobic and anaerobic treatment conditions. Azoreductases are present in a wide range of bacteria, including *Pseudomonas sp., Enterococcus faecalis YZ 66* and *Bacillus* sp. [52–54]. The azo bond of Basic Orange 2 dye has been reduced and converted to two substituted benzene derivatives, aniline and 3, 4-diaminobezeminium chloride, by the action of the bacterial enzyme azoreductase. Furthermore, aniline converts to a benzene ring by the deamination process with the help of the deaminase enzyme of the bacterial system. As methyl donors S-adenosyl-methionine and methyltetrahydrofolate are already present in bacteria and have been reported to be involved in the methylation of benzene ring [55], subsequent methylation occurs, and benzene converts to o-xylene. The other part of the degraded dye across the azo linkage, 3, 4-diaminobezeminium chloride, is converted to a benzene ring by enzymatic action through subsequent deamination reactions. After deamination, the methylation of the benzene ring transformed benzene to p-xylene in the final step. Figure 16 shows the proposed mechanism for the biodegradation of Basic Orange 2 dye by *Escherichia coli.*

**Figure 16.** Proposed mechanism for the degradation of Basic Orange 2 by *Escherichia coli.*

## 4. Conclusions

The aim of this study was to assess *Escherichia coli's* biodegradation potential for the azo dye Basic Orange 2. The chosen bacterial strain degraded Basic Orange 2 more effectively compared to the other tested strains. The effect of physicochemical parameters on bacterial degrading efficiency was investigated, whereby using 3-day intervals, a temperature of 40 °C, pH 7, 20 ppm dye concentration, 1666 mg/L glucose supplementation, 666 mg/L NaCl salt concentration, 1000 mg/L urea concentration and sodium benzoate as a redox mediator at concentration of 66 mg/L were found to be the best conditions to achieve maximum decolorization. All these optimal conditions were then combined into a single experiment, and as a result, 89.88% degradation of the selected dye was achieved. The effect of the four process variables of pH, dye concentration, incubation time and temperature significantly influenced the degradation of the dye at ($p < 0.05$), as depicted from the result of the ANOVA of the response surface methodology (RSM). The spectroscopic techniques UV/Vis, FTIR and GCMS were used to study the degradation process and the metabolites formed under the optimal conditions. From the GC-MS pattern, it was concluded that the bacteria enzymatically degraded the corresponding dye to o-xylene and p-xylene. As a proposed mechanism, it was presumed that the bacterial enzyme azoreductase initially breaks the azo linkage of Basic Orange 2 dye, which is followed by deamination and reduction of the aniline ring to benzene rings. Subsequent methylation of both benzene rings converts them to o-xylene and p-xylene. From the results, it was concluded that *Escherichia coli* could be effectively used in the reclamation of dye-loaded water as an effective and efficient strain for the bioremediation of textile wastewater containing Basic Orange 2 dye. However, further research is needed to enhance the degradation capability of the selected bacteria through variation of the experimental conditions. Furthermore, at this stage, it is too early to propose a mechanism, and other metabolites may also be formed during the process; therefore, further experiments in this area are also required.

**Author Contributions:** M.I. conducted the research work and wrote the paper. M.Z., M.N. and M.M.H. supervised the work and revised the paper. Formal analysis were performed by R.U., D.A.A.F., M.S.E. and N.G. I.Z. helped in the investigation. A.A.O. helped in the design of the response surface methodology (RSM) for optimization. All authors have read and agreed to the published version of the manuscript.

**Funding:** The authors extend their appreciation to the research support, project number (RSP-2021/190), from the King Saud University, Riyadh, Saudi Arabia.

**Institutional Review Board Statement:** Not applicable.

**Informed Consent Statement:** Not applicable.

**Data Availability Statement:** Not applicable.

**Conflicts of Interest:** The authors declare no conflict of interest. The funders had no involvement in the study's design, manuscript writing, data collection, analysis or interpretation, or the decision to publish results.

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
