# Peer review of "Biological Degradation of the Azo Dye Basic Orange 2 by Escherichia coli: A Sustainable and Ecofriendly Approach for the Treatment of Textile Wastewater"

_water, doi:10.3390/w14132063_

Round 1

Reviewer 1 Report

Material and methods section. Since the dye is not a natural product I guess that E. coli needed an acclimatization period to achieve proper degradation kinetics. I cannot see this point in the manuscript. Please, clarify.

Lines 65-66. Most of the cited treatments do not produce "large amounts of sludge". Maybe chemical precipitation is the one with the highest sludge production. Please, cite which of the mentioned treatments would generate more sludge.

Line 95: "has" should be removed.

Line 124: please correct "0.04 gm/L"

Line 125: Why did you choose 40 ppm? Is it related to the dye concentration in textile wastewater? Please justify.

Line 151: please correct "Dedegradation" and "Initialabsorbance" on equation 1.

Line 237: "decolonization"??

Line 241: A "3 day incubation duration" time is a rather long one. In activated sludge treatment a typical hydraulic residence time is about 6-12 hours. Do you think that the obtained kinetics are rapid enough to be applied in the real world? Please comment.

The experimental section also provides results. Experimental and results should be separated.

Lines 288-289: the English should be revised.

Section 3.2.4. Lines 319-321. Did you measure the bacterial concentrations or mass? Why shorter incubation times than 3 days were not measured? 3 days is rather a long time.

Section 3.2.5. The concentration of glucose and the other nutrients should be expressed in a more normalized fashion, such as mg/L. What happened at higher glucose concentrations? There is an increasing trend in the removal. 

Line 299. " and the optimal pH range being 6 to 10": optimal for degradation?

Line 364: Please correct "a 94.41 0.98 percent yield" and "o 79.35 1.26".

Line 367: " on Dye Basic Orange 2 % degradation." Please clarify this "2 %"

Figure 11 should be re-arranged: there is no need to have so many decimals in degradation % and the vertical label is covered by the numbers. The same applies to Fig. 12.

Section 3.4 is too long. It should be reduced, particularly the theoretical aspects.

I wonder if Fig. 14 is necessary. Please consider removal. Only the xylenes were detected? If so, how can you propose a degradation mechanism with so little experimental evidence? Please clarify.

Author Response

Reviewer 1

Comment: Material and methods section. Since the dye is not a natural product I guess that E. coli needed an acclimatization period to achieve proper degradation kinetics. I cannot see this point in the manuscript. Please, clarify.

Response: Thank you worthy reviewer, without acclimation this could not happen, and if see the experimental duration is 72 h. During this period first it is acclimated. Also, bacteria need growth media and other nutritional requirements such as C, N etc, proper degradation is expected to begin when bacteria is alive and active. Bacteria enzymes are responsible in degradation kinetics.  In our study acclimatization period began after 6-12 hours. The maximum degradation occurred at 72 hours. After this time, degradation decreased which is largely due to bacterial enzymes saturation and decrease in growth.

Comment: Lines 65-66. Most of the cited treatments do not produce "large amounts of sludge". Maybe chemical precipitation is the one with the highest sludge production. Please, cite which of the mentioned treatments would generate more sludge.

Response: Treatment method which generates more sludge has been cited accordingly in the respective section.

Comment: Line 95: "has" should be removed.

Response: The mistake has been corrected .Thanks.

Comment: Line 124: please correct "0.04 gm/L"

Response: The mistake has been corrected.

Comment: Line 125: Why did you choose 40 ppm? Is it related to the dye concentration in textile wastewater? Please justify.

Response: worthy reviewer, scientific studies are performed with a range of concentrations rather than a single concentration. This range include high and low levels. Mostly lower levels tested are related with dyes concentration in wastewaters. The high levels are evaluated for toxic effects whereas the lower one is evaluated to estimate the efficiency of the applied technique. Dye at high concentration is toxic and cause bacterial death. The presence of dye in the receiving water and environment is considered hazardous even at very low concentration. We have carried out study using a range of concentration to determine the effect of degradation of the dye using E. coli. Similar studies have conducted their investigation using the range of the dye concentration for the degradation of dye using bacteria strain. Some cited works are as follows:

  1. Khan, A. U., Rehman, M. U., Zahoor, M., Shah, A. B., & Zekker, I. (2021). Biodegradation of brown 706 dye by bacterial strain pseudomonas aeruginosa. Water13(21), 2959.
  2. Khan, A. U., Zahoor, M., Rehman, M. U., Shah, A. B., Zekker, I., Khan, F. A. & Mohamed, H. R. (2022). Biological Mineralization of Methyl Orange by Pseudomonas aeruginosa. Water14(10), 1551.
  3. ARIFFIN, B.K. and ARIFFIN, F., 2020. BIODEGRADATION OF AZO DYE (REACTIVE GREEN 19) BY Pseudomonas aeruginosa ISOLATED FROM TEXTILE EFFLUENT. Malaysian Applied Biology49(4), pp.1-8.

Comment: Line 151: please correct "Dedegradation" and "Initiala bsorbance" on equation 1.

Response: These mistakes have been corrected accordingly.

Comment: Line 237: "decolonization"??

Response: The spelling mistake has been corrected.

Comment: Line 241: A "3 day incubation duration" time is a rather long one. In activated sludge treatment a typical hydraulic residence time is about 6-12 hours. Do you think that the obtained kinetics are rapid enough to be applied in the real world? Please comment.     

Response: 3 days is optimum time to degrade the dye. Degradation may start 6-12 h after acclimation period but in 3 days maximum deodorization of the dye occured. Moreover in the activated sludge the concentration of microbial colonies and growth is more as compared to test tube experiment. According to the cited work given below 3 days is optimum incubation duration for bacteria to degrade the given dye.

  1. Khan, A. U., Rehman, M. U., Zahoor, M., Shah, A. B., & Zekker, I. (2021). Biodegradation of brown 706 dye by bacterial strain pseudomonas aeruginosa. Water13(21), 2959.
  2. Khan, A. U., Zahoor, M., Rehman, M. U., Shah, A. B., Zekker, I., Khan, F. A. & Mohamed, H. R. (2022). Biological Mineralization of Methyl Orange by Pseudomonas aeruginosa. Water14(10), 1551.

Comment: The experimental section also provides results. Experimental and results should be separated.

Response: The results have been removed from experimental section and placed in results section accordingly.

Comment: Lines 288-289: the English should be revised.

Response: This sentence has been revised accordingly.

Comment: Section 3.2.4. Lines 319-321. Did you measure the bacterial concentrations or mass? Why shorter incubation times than 3 days were not measured? 3 days is rather a long time.

Response: Worth reviewer, we have observed the dyes color visually and after 72 hour no appreciable changes were observed, therefore this was taken as optimum time. We have not measure the bacterial rather our focus was on color change rather the bacterial mass.

Comment: Section 3.2.5. The concentration of glucose and the other nutrients should be expressed in a more normalized fashion, such as mg/L. What happened at higher glucose concentrations? There is an increasing trend in the removal. 

Response: Since we conducted experiments in test tube and its capacity is 15 mL that is why we expressed the concentration in mg/15 mL. At higher glucose concentration the removal efficiency increases because glucose act as carbon source for bacteria and hence increase its growth which leads to high mass production needed for the degradation. That is why high degradation efficiency has been observed.

Comment: Line 299. " and the optimal pH range being 6 to 10": optimal for degradation?

Response: Extreme acidic and alkaline conditions affects bacterial growth and decrease removal efficiency, therefore the pH range of 6-10 was used in our study. Accordingly, related works is as follows;

  1. Saratale, R.G.; Saratale, G.D.; Chang, J.S.; Govindwar, S.P. Bacterial decolorization and degradation of azo dyes: A review. Taiwan. Inst. Chem. Eng.2011, 42(1), 138–157.
  2. Khan, A. U., Rehman, M. U., Zahoor, M., Shah, A. B., & Zekker, I. (2021). Biodegradation of brown 706 dye by bacterial strain pseudomonas aeruginosa. Water13(21), 2959.
  3. Khan, A. U., Zahoor, M., Rehman, M. U., Shah, A. B., Zekker, I., Khan, F. A. & Mohamed, H. R. (2022). Biological Mineralization of Methyl Orange by Pseudomonas aeruginosa. Water14(10), 1551.

Comment: Line 364: Please correct "a 94.41 0.98 percent yield" and "o 79.35 1.26".

Response: These mistakes has corrected.

Comment: Line 367: " on Dye Basic Orange 2 % degradation." Please clarify this "2 %"

Response: The % degradation has been separated from the dye name (Basic Orange 2). Thanks.

Comment: Figure 11 should be re-arranged: there is no need to have so many decimals in degradation % and the vertical label is covered by the numbers. The same applies to Fig. 12.

Response: This has been revised accordingly.

Comment: Section 3.4 is too long. It should be reduced, particularly the theoretical aspects.

Response: Section 3.4 has been reduced accordingly.

Comment: I wonder if Fig. 14 is necessary. Please consider removal. Only the xylenes were detected? If so, how can you propose a degradation mechanism with so little experimental evidence? Please clarify.

Response: According to GCMS results a number of compounds were detected. Some were impurities in solvents whereas some were from dyes origin. Among the detected compounds the close relation of xylene was found with dye structure. That is why the claim has been made.

Reviewer 2 Report

1. Abstract: Written well but general terms must be removed and just results should be discussed.

2. Introduction: Why necessary to do this research must be provided with  Latest literature comparison must be provided, references aren't updated.

3. Methodology: Sections must be reduced in methodology section e.g. section 2.14, 2.15 and 2.16 can be made one section. How authors carried out degradation experiment is not provide so must be added.

4. Results are discussed well but there must be visual experimental pictures (like microbial growth images SEM, culture images and time scale analysis of dye degradation). For SEM sample preparation of microbes authors can refer this latest paper https://doi.org/10.1016/j.cej.2021.134361 and cite in manuscript.

5. Conclusions: must be revised and add some conclusive remarks along with some future perspective.

Author Response

Reviewer 2

  1. Comment: Abstract: Written well but general terms must be removed and just results should be discussed.

Response: The abstract has been corrected accordingly.

  1. Comment: Introduction: Why necessary to do this research must be provided with Latest literature comparison must be provided, references aren't updated.

Response: The degradation of dye has extensively been reported using other physico-chemical methods in the literature. However, the use of bacterial degradation of dye has rarely been reported. This study reports for the first time the biodegradation of basic orange 2 dye using E. coli. Also, the study critically examined the effect of process conditions both in one time factor optimization and optimization using central composite design to determine the effect of the interaction of the operational parameters on the degradation efficiency of E. coli. This has been added in page 3.

  1. Comment: Methodology: Sections must be reduced in methodology section e.g. section 2.14, 2.15 and 2.16 can be made one section. How authors carried out degradation experiment is not provide so must be added.

Response: Section 2.14, 2.15 and 2.16 has combined to single section. The degradation experiment has been described in degradation/decolorization activity in methodology section.

  1. Comment: Results are discussed well but there must be visual experimental pictures (like microbial growth images SEM, culture images and time scale analysis of dye degradation). For SEM sample preparation of microbes authors can refer this latest paper https://doi.org/10.1016/j.cej.2021.134361 and cite in manuscript.

Response: Images has already been added. Few new are added. The given reference was cited.

  1. Comment: Conclusions: must be revised and add some conclusive remarks along with some future perspective.

Response: Conclusive remarks along with future perspective have been added in the revised manuscript.

Reviewer 3 Report

The current study (ID: water-1773017) discusses an interesting point in the field of bioremediation, while significant major corrections are necessary before the publication can be accepted. Keywords need to enrich with some related valuable words. The introduction is inadequate and need some improvements and enriched with some recent articles explain the harmful impact of toxic dye on the aquatic habitats and the importance of trend of the bioremediation, in general, maybe use the following articles (https://doi.org/10.3390/ma15113922 ; https://doi.org/10.3390/polym14071375 ). Some important characterizations may be useful in the explanation of the current data such as Brunauer–Emmett–Teller surface area analysis (BET). In all the manuscript parts, authors must carefully revise the units of subscript and/or superscript. There are too many mistakes that must be corrected. Authors must add the standard division (SD) values on the Figs from numbers 2 to 10. Fig 2 and 14 need to change with others of high resolution. Moreover, the Figure with the caption “GC chromatogram of Basic Orange 2 dye after bacterial degradation” must take Fig number. Also, correct the number of Figures with the caption “Proposed mechanism for …..” Many paragraphs need to be re-written and the English language and style need to improve.

Author Response

Reviewer 3

Comment: The current study (ID: water-1773017) discusses an interesting point in the field of bioremediation.

Response: Thanks referee for your esteemed remarks and appreciation of our works.

Comment: Keywords need to enrich with some related valuable words.

Response: New keywords has been added.  

Comment: The introduction is inadequate and need some improvements and enriched with some recent articles explain the harmful impact of toxic dye on the aquatic habitats and the importance of trend of the bioremediation, in general, maybe use the following articles (https://doi.org/10.3390/ma15113922 ; https://doi.org/10.3390/polym14071375 ).

Response: The introduction has been improved. These articles and other recent articles has enhanced the manuscript. The proposed papers were cited accordingly.

Comment: Some important characterizations may be useful in the explanation of the current data such as Brunauer–Emmett–Teller surface area analysis (BET).

Response: Worthy reviewer, we are working on bacterial strains here not on adsorbent to provided BET surface area. For bacterial cultures no such experiment is possible to subject them to Nitrogen adsorption desorption tests. We have carried out characterization using FT-IR to examine functional group changes. GC-MS was utilized for the identification of dye degraded products or metabolites. This investigations’ has been carried in our study which aims to carry out identification and isolation of dye metabolites as it relates to its degradation.

Comment: In all the manuscript parts, authors must carefully revise the units of subscript and/or superscript.

Response: The manuscript the units of subscript and superscript were revised accordingly.

Comment: There are too many mistakes that must be corrected. Authors must add the standard division (SD) values on the Figs from numbers 2 to 10.

Response: This has been added in the Figures.

Comment: Fig 2 and 14 need to change with others of high resolution.

Response: The figures has been enhanced.

Comment:  Moreover, the Figure with the caption “GC chromatogram of Basic Orange 2 dye after bacterial degradation” must take Fig number.

Response: Figure number has been added to “GC chromatogram of Basic Orange 2 dye after bacterial degradation.

Comment: Also, correct the number of Figures with the caption “Proposed mechanism for …..” Many paragraphs need to be re-written and the English language and style need to improve.

Response: The number of Figures with the caption “Proposed mechanism for …..” has been corrected. The manuscript has been corrected for English editing by a native speaker.

Round 2

Reviewer 1 Report

The authors have not followed any of my advices, examples:

- selecting 40 ppm as study concentrations has not been justified

- concentrations of urea, glucose, etc., should be expressed in mg/L or ppm not in mg/15 mL

- section 3.4 is too long, it has not been shorten,

- Figures 14 are not necessary,

- no mention on the acclimatization period is made in the text

Author Response

Reviewer 1

The authors have not followed any of my advices, examples:

Reviewer: selecting 40 ppm as study concentrations has not been justified

  • Response: Worthy reviewer, in our preliminary studies the dye concentration above 40 ppm was toxic. So increase in the dye concentration above 40 ppm causes bacterial death which thus was not considered as it was toxic for the bacterial colonies. A relevant study in literature also reveals this fact. (http://s-o-i.org/1.15/ijarbs-2016-3-3-27)  ,  (  https://doi.org/10.3390/w13212959)
  • Another reason is when concentration of dye increases above 40 ppm then the absorbance measurement in UV-Visible spectrophotometer is difficult. UV-Visible spectrophotometer shows accurate absorbance when concentration is low. At high dye concentration the absorbance value reaches from 3-4 and hence it is difficult to record the absorbance. In such cases the concentrations measured are erroneous and one should always keep concentration for which the absorbance is below 1.

Reviewer: - Concentration of urea, glucose, etc., should be expressed in mg/L or ppm not in mg/15 mL

  • Response: The concentrations of urea, glucose, sodium chloride and redox mediators have been expressed in mg/L in the revised manuscript.

Reviewer: - section 3.4 is too long, it has not been shortened.

  • Response: The text has been shortened accordingly.

- Figures 14 are not necessary,

  • Response: Worthy reviewer these are figures of GC and GC-MS chromatograms of the Dye metabolites. Our study is mainly focused on identification of Dye metabolites. Without these figures we cannot fully justify our GC-MS results. According to our study model these figures are needed. In other related articles such type of figures has been mentioned by authors; (https://doi.org/10.3390/w14101551), (  https://doi.org/10.3390/w13212959)

Reviewer: - no mention on the acclimatization period is made in the text

  • Response: The acclimatization period has been added in page 10 of the revised manuscript.

Reviewer 2 Report

Authors have revised manuscript very well according to suggestion.

Author Response

Reviewer 2

Authors have revised manuscript very well according to suggestion.

  • Response: Your valuable suggestions have improved the quality and scope of our manuscript. Thank you

Reviewer 3 Report

The paper is accepted although There are some modifications still needed:

1. The qualities of all Figs need to improve, especially Fig 1, 14a, and 15.

2. authors forget the word "Figure 12"

3. Merge the captures of Fig 14(a) and Fig 14(b)

Author Response

Reviewer 3

Reviewer: The paper is accepted although There are some modifications still needed:

Response: Your valuable suggestions are welcomed. Thank you

Reviewer: 1. The qualities of all Figs need to improve, especially Fig 1, 14a, and 15.

  • Response: The quality of the mentioned figures has been improved accordingly.

Reviewer: 2. authors forget the word "Figure 12"

  • Response: This mistake has been corrected. Thank you

Reviewer:  3. Merge the captures of Fig 14(a) and Fig 14(b)

  • Response: The caption has been merged accordingly.